# Who has to tell their trauma story and how hard will it be? Influence of cultural stigma and narrative redemption on the storying of sexual violence

**Brianna C. Delker**[1]*, **Rowan Salton**[1], **Kate C. McLean**[1], **Moin Syed**[2]

**1** Department of Psychology, Western Washington University, Bellingham, Washington, United States of America, **2** Department of Psychology, University of Minnesota, Minneapolis, Minnesota, United States of America

* Brianna.Delker@wwu.edu

**Data Availability Statement:** The data underlying the results presented in the study are available from Open Science Framework, https://osf.io/qj8f4/

## Abstract

Although survivors of sexual violence have shared their stories with the public on social media and mass media platforms in growing numbers, less is known about how general audiences perceive such trauma stories. These perceptions can have profound consequences for survivor mental health. In the present experimental, vignette-based studies, we anticipated that cultural stigma surrounding sexual violence and cultural preference for positive (redemptive) endings to adversity in the United States (U.S.) would shape perceptions. Four samples of U.S. adults ($N$ = 1872) rated first-person narratives of 6 *more stigmatizing* (i.e., sexual violence) or *less stigmatizing* (e.g., natural disaster) traumatic events. Confirming pre-registered hypotheses, sexual violence trauma (versus other types of trauma) stories were perceived as more difficult to tell, and their storytellers less likeable, even when they had redemptive endings. Disconfirming other pre-registered hypotheses, redemptive (versus negative) story endings did not boost the perceived likelihood or obligation to share a sexual violence trauma story. Rather, redemptive (versus negative) story endings only boosted the perceived likelihood, obligation, and ease of telling other, less stigmatizing types of trauma stories. Findings suggest that sexual violence survivors do not benefit, to the same degree as other survivors, from telling their stories with the culturally valued narrative template of redemption. Clinical and societal implications of the less receptive climate for sexual violence stories are discussed.

## Introduction

In the era of the Me Too Movement, sexual violence survivors are coming forward to share stories of trauma widely on social media and mass media platforms. The degree to which mainstream audiences are receptive to such stories of trauma has profound consequences for individuals and society. These stories have the potential to change the course of careers and businesses, and to impact major judicial appointments and political elections. As an

**Funding:** BCD and KCM received a research grant from Western Washington University's Center for Cross-Cultural Research (#CRC_Red) and a Manuscript Preparation Grant from Western Washington University's Office of Research and Sponsored Programs (#MF1855). The funders did not play any role in the study design, data collection and analysis, decision to publish, or preparation of the manuscript.

**Competing interests:** The authors have declared that no competing interests exist.

international social justice movement and social media hashtag for survivors seeking to break the silence around sexual violence and harassment, #MeToo has created an unprecedented audience for interpersonal violence stories. However, surprisingly little is known about how general audiences perceive the trauma story-telling process, such as the perceived difficulty or obligation to tell such stories. Given that stigmatizing social responses to interpersonal violence disclosures are connected to worse mental health outcomes for survivors and less help-seeking [1,2], better understanding such perceptions may ultimately help promote survivors' well-being.

The present study examines the degree to which public perceptions of trauma story-telling are shaped by *cultural preference* for positive endings to stories of adversity, and *cultural stigma* surrounding sexual violence. Recent research shows a strong cultural preference in the United States for stories of trauma to be *redeemed*—to conclude positively with strength gained or lessons learned [3]. Cultural celebration of redemption may create a welcoming reception for trauma stories with redemptive endings, perhaps even a perceived obligation to tell these stories. But importantly, not all trauma survivors are regarded with equal goodwill. Relative to traumas caused naturally (e.g., natural disaster) or by people accidentally (e.g., vehicle collision), sexual assault and other forms of interpersonal violence caused by people intentionally are regarded as less legitimate traumas, and the survivors stigmatized as more blameworthy, less credible, and more personally flawed [2]. We designed the present vignette-based experimental study to examine the ways that trauma story *type* (sexual violence versus other types of trauma) and *ending* (positive versus negative) interact to shape perceptions of survivor stories. Perceptions assessed include survivors' likelihood of sharing their experience, their obligation to share, the difficulty and burden of doing so, and the likability of the storyteller. We selected these measures of perception due to their clinical and practical relevance, unpacked further below.

## Why "cultural" preference and stigma?

By the term *culture*, we refer broadly to mainstream, dominant U.S. cultural values constructed and normalized against the backdrop of settler-colonialism, white supremacy, patriarchy, and consumer capitalism. Arguably, dominant cultural values in the U.S. include rugged individualism, personal responsibility, personal "grit" and stoicism, and presumptions of a meritocratic and "just world" for all [4,5,6]. To the advantage of powerful groups, dominant cultural values are made to seem normal, desirable, and universal, despite enormous social and economic inequality in the U.S. As such, we use the term "culture" with the intention to evoke its critical and contested dimensions: both culture and the subjectivity of persons are constructed within the setting of unequal power relationships [7]. At times, individuals may accommodate to oppressive understandings of the world in their values and claims to identity, and at times they may resist them [8,6]. By subsuming many racial and ethnic groups and identities under the sweeping umbrella of one U.S. culture, for the purposes of this work, we wish to make clear that we do not want to further marginalize any group or to wash out important differences between and within groups. Rather, we aim to consider the degree to which dominant cultural values make claims on us all. When considering the implications of our findings, we argue for the importance of future research addressing the degree to which trauma story-telling perceptions differ between and among groups on the basis of gender, race, ethnicity, and other dimensions of identity.

## Cultural stigma surrounding the experience of sexual violence informs reactions to disclosure

*Cultural stigma* legitimizes sexual violence and devalues survivors by culturally constructing the reality of the event in ways that minimize its severity [9,10]. Individuals and institutions

[9] and cultural products (e.g., media, internet content, advertising; [11]) can be vehicles of cultural stigma. The majority of stigma research with interpersonal violence survivors has focused on survivors' internalized sense of shame and self-blame attached to their diminished status [2]. However, trauma researchers and clinicians have long been sensitive to how social reactions to survivor disclosures can constitute a mix of positive and negative reactions, the latter of which can constitute stigmatizing reactions and can have profound consequences for survivor well-being [1,12–14]. For example, children and adults who disclose experiences of sexual assault are often met with stigmatizing reactions such as disbelief, victim blame, and shaming, such as implying that the survivor is "tainted" by the experience [2,15–19,20 p. 1746].

In the setting of dominant cultural values around personal responsibility and belief in a safe, just world, disbelief and victim-blaming responses to stories of sexual violence may allow bystanders to preserve the comforting assumption that acquaintances, friends, family, colleagues, and respected public figures could not be perpetrators. By contrast, disbelief and victim-blame are arguably less pronounced, on average, in response to stories of other types of trauma like natural disasters, life-threatening illness, or sudden traumatic loss of a loved one (though see [21] and [22] for exceptions). Traumas such as natural disasters and their effects are also by their nature more visible and public than acts of sexual violence, which tend to be committed behind closed doors. Given the shame and negative attributions attached to sexual violence, we conceptualize such experiences as *more* stigmatizing traumas, distinguishing between these and other, *less* stigmatizing traumas. We return to the distinction between these trauma types after addressing the broad cultural preference for redemptive stories.

## The cultural preference for redemptive stories may not apply equally to survivors of sexual violence

Mainstream media and politics in the United States celebrate stories of protagonists who transcend adversity and find redemption in the aftermath of trauma [4,23]. In a redemptive story, negative events are followed by positives such as gratitude, success, strength gained, or lessons learned. Not surprisingly, American audiences like redemptive stories and their storytellers. Recent empirical work shows that adults in the U.S. rate redemptive trauma stories as more personally preferable, and trauma survivors who tell redemptive stories as having more adaptive, culturally valued traits (e.g., conscientiousness, emotional stability [3]). As such, those who can tell a redemptive story communicate not only what happened but also a virtuous image of self—someone who is agentic and positive, valued traits in dominant U.S. culture. In this work, we use the terms *story* and *storytelling* intentionally: a story is a de facto disclosure, but it is more than that—it has a narrative structure and it says something (intentionally or not) about who the storyteller is.

At first glance, the strong preference for redemptive stories might seem to present a culturally valued template (or, *master narrative*; [3,23–25]) for sexual violence survivors to publicly share their stories with confidence that audiences will be receptive. However, despite the clear U.S. preference for redemptive stories over ones with negative endings, it is not yet known whether this preference extends to stories of sexual trauma, a story with particular challenges for sharing. For example, do audiences perceive survivors of sexual violence as any more *likeable* when they tell stories with redemptive endings, versus stories without redemptive endings? Do audiences perceive stories of sexual traumas with redemptive endings as any more *likely* to be shared or any *easier* for survivors to share than such stories with negative endings? These questions should not only be of public interest and concern in the era of #MeToo, but they have crucial implications for survivor well-being and willingness to disclose their

experiences. Although we do not measure the latter constructs in this study, we turn to these issues here due to their relevance.

## The proposed paradox of sexual violence survivors being less likely and more obligated to share trauma stories with redemptive endings

Children and adults who have experienced physical, psychological, or sexual violence tend to disclose or report their experiences at low rates, meaning that stories of these survivors are, in effect, silenced [26–28]. When stories are silenced, victims are less likely to access resources in the health and legal systems, deferring justice and healing [29]. Even sexual assault survivors who have recovered from the life disruption of trauma may not feel comfortable sharing their history with others. Their reluctance may relate to the stigma attached to sexual violence, or to the perception that their stories will be emotionally difficult for others to hear, even for trained professionals such as therapists [30]. Sex and sexuality, in general, are sensitive and even taboo topics among many cultural and religious groups, heightening the difficulty of sharing about sexual traumas, even when their emotional intensity has abated. Consistent with empirical research on low rates of disclosure, we anticipate that the audiences in our study will perceive stories of sexual violence as more emotionally difficult to share and less likely to be shared (relative to stories of other types of trauma), *even when the endings are redemptive*. Thus, we argue that the pronounced cultural stigma attached to sexual violence will counteract and silence any opportunity that was created by a redemptive trauma story.

However, we also anticipate that audiences will, paradoxically, perceive survivors of sexual violence traumas with redemptive (versus negative) endings as *more obligated* to share their stories. People who have experienced the degradation of sexual violence and yet found strength in its aftermath may be expected to open up about their experiences to disrupt stigma and potentially benefit and empower other victims. This has been a positive cultural shift heralded by the Me Too movement, which provides a platform for showing other victims that they are not alone, that thriving after trauma is possible. In the event that a survivor with a redemptive story comes forward in this way, this person has assumed what we call a *survivor identity*, a type of redemption characterized by service to others. At the same time, in a climate where institutions and individuals accused of wrong-doing continue to resist accountability, it is often left up to individual survivors—more and more voices, louder and louder—to ensure that perpetrators are held accountable. The perception that survivors of sexual violence must bear the burden to tell their stories is consistent with the implicit cultural assumption that victims are personally responsible for what happens to them—and what happens next.

Overall, we propose that there is a paradoxical set of dominant cultural expectations for sexual violence survivors: to be silent about their experiences *and* to tell others about their experiences, if they can tell a story with a happy, and empowered, ending.

## The present study

To test this claim we conducted an experimental, vignette-based study, in which we developed a set of written, first-person trauma narratives to which participants were randomly assigned [3]. In each trauma story, a protagonist describes experiencing a traumatic event that is a *more stigmatizing trauma* (childhood sexual abuse or adult sexual assault) or *other, less stigmatizing type of trauma* (trauma caused naturally or by people accidentally- car accident, hurricane, childhood life-threatening illness, or traumatic loss- sudden death of a friend). Each trauma story concludes with one of three possible endings: a *negative* ending in which the storyteller is still suffering in the aftermath of the trauma; a *redemptive* ending in which the storyteller has personally grown or gained from the experience; or a *survivor identity* ending, a type of

redemptive ending in which the storyteller has been inspired to serve others who have experienced similar traumas. We included the survivor identity ending in order to ensure adequate content validity for the redemptive story-telling construct, as redemption can include not only personal growth but also service to others, as exemplified by some of the survivor-advocates who have told their stories publicly in the Me Too movement.

**Hypotheses.**   Our hypotheses follow. The primary hypotheses were pre-registered on the Open Science Framework (OSF; [31]). Exploratory analyses are clearly marked as such in the text. All study materials, de-identified datasets, and data analysis code can be found on OSF [31].

The first set of hypotheses compare sexual violence trauma stories to other, less-stigmatizing trauma stories, separately by ending (negative, redemptive, or survivor identity). We predict that sexual violence stories with negative endings will be perceived as *more difficult to share*, *more burdensome to share*, *less obligatory to share*, and *less likely to be shared* than other types of trauma stories with negative endings (H1). The direction of our hypotheses for stories with redemptive (H2) and survivor identity (H3) endings are the same, except we predict that sexual violence stories with these positive endings will be perceived as *more obligatory to share* and *more likely to be shared* than less-stigmatizing stories with such positive endings.

The next set of hypotheses compare negative, redemptive, and survivor identity stories, examining trauma type (sexual violence versus other) as a moderator of the effect of story ending on perceptions. We predict that trauma stories with negative endings will be perceived as *more difficult to share*, *more burdensome to share*, *less obligatory to share*, and *less likely to be shared* than trauma stories with redemptive and survivor identity endings (H4). Further, we propose that trauma type (more versus less stigmatizing) will moderate the proposed associations in Hypothesis 4, such that, in general, the switch from negative to positive endings would create more perceived obligation, burden, and likelihood to tell sexual violence stories, but *not* make sexual violence stories seem any easier to tell (H5).

The final set of analyses were exploratory and evaluate the perceived likeability of storytellers based on trauma type and ending. For these exploratory analyses, we anticipate that participants will evaluate the narrators of sexual violence trauma stories as less likeable than the narrators of other types of trauma stories, whether the story endings are negative (E6) or positive (redemptive or survivor ending) (E7). These are exploratory analyses due to the absence of prior research on how the perceived likeability of trauma storytellers varies by trauma type. We propose these exploratory analyses due to the relevance of perceived likeability for the support and encouragement that survivors can expect to receive from mainstream audiences. Although redemptive storytellers are perceived as more likeable than storytellers with negative endings, averaged across trauma type [3], it is not yet known whether the redemptive boost in likeability extends to sexual violence survivors.

We anticipate that the hypothesized effects above will persist even when controlling for personal characteristics connected to more stigmatizing traumas and perceptions of these traumas in prior research (participant gender, participant lifetime trauma history; [32–35]). These variables were included as pre-registered covariates in the analysis, but we did not specify nor test specific directional hypotheses about the covariates. Although it would be reasonable to wonder whether participant gender and trauma history might serve as moderators of the proposed effects of story type on perceptions, we leave these questions to future research. In the Discussion section, we consider the limitations of this pre-registered decision. Finally, in order to determine the robustness of effects, we recruited adult participants across a range of platforms and we examined story ratings both between and within subjects.

## Method

Study procedures and materials were identical for the mixed within- and between-subjects experimental design (Study 1) and within-subjects experimental design (Study 2) created to test our hypotheses. The exception was a different procedure for random assignment to conditions, explained further below. Unless otherwise stated, the information in the following Method subsections pertains to both studies.

### Participants

Participants ($N = 1,872$) were adults at four U.S. sites: Amazon's Mechanical Turk (MTurk; $n = 336$; Study 1), Qualtrics online panels ($n = 190$; Study 2), and two public universities in distinct geographic locations (University 1, $n = 664$; University 2, $n = 682$). Overall, the adults in these samples were predominantly female and White with an average age in the late 20s overall (range 18–82). For a full breakdown of demographics please refer to Table 1.

### Procedure

The Research Compliance Organizations of Western Washington University and the University of Minnesota approved the study protocol. Online surveys were administered on Qualtrics.com. Participants on both MTurk and Qualtrics online panels signed up to participate in the study titled "Evaluating Stories" for financial incentive upon valid completion of the surveys. Students from the two universities were recruited similarly using university-based online research management systems, however they received course credit for study participation (alternative avenues for course credit were available). All participants provided informed consent electronically when they agreed to participate.

**Table 1. Demographic characteristics of study 1 & 2 participants.**

| Characteristic | Total Sample ($N = 1,872$) | MTurk ($n = 336$) | University 1 ($n = 664$) | University 2 ($n = 682$) | Qualtrics ($n = 190$) |
|---|---|---|---|---|---|
| | $N$ (%) | $n$ (%) | $n$ (%) | $n$ (%) | $n$ (%) |
| Gender | | | | | |
| female | 1280 (68.4) | 181 (53.9) | 497 (74.8) | 454 (66.6) | 148 (77.9) |
| male | 570 (30.4) | 154 (45.8) | 153 (23) | 224 (32.8) | 39 (20.5) |
| non-binary | 14 (0.7) | NA | 11 (1.6) | 3 (0.4) | NA |
| transgender | 8 (0.4) | 1 (0.3) | 3 (0.45) | 1 (0.1) | 3 (1.6) |
| Race | | | | | |
| White | 1478 (79.0) | 275 (81.8) | 553 (83.2) | 498 (71.7) | 152 (80.0) |
| Black | 100 (5.3) | 32 (9.5) | 17 (2.5) | 36 (5.3) | 15 (7.9) |
| Asian | 240 (12.8) | 22 (6.5) | 77 (11.6) | 134 (19.6) | 7 (3.7) |
| Native Hawaiian / Pacific Islander | 17 (0.9) | 5 (1.5) | 10 (1.5) | 1 (0.1) | 1 (0.5) |
| Native American / Alaska Native | 18 (0.96) | 4 (1.2) | 8 (1.2) | 3 (0.4) | 3 (1.6) |
| Latinx | 112 (6.0) | 11 (3.3) | 58 (8.7) | 34 (5.0) | 9 (4.7) |
| **** | 31 (1.7) | 4 (1.2) | 11 (1.7) | 14 (2.1) | 2 (1.0) |
| | $M$ (range) | $M$ (range) | $M$ (range) | $M$ (range) | $M$ (range) |
| Age | 27.0 (18–82) | 41.38 (21–71) | 20.13 (18–67) | 19.96 (18–42) | 50.89 (18–82) |

**** denotes participants who provided a race not represented by the available categories, or two or more races. Total $n$(%) of race items do not add up to 100% because participants could select more than one. Table reprinted with permission from [3].

This experimental vignette-based online study used a 6 (*story*: sexual violence trauma [*k* = 2], other type of trauma [*k* = 4]) x 3 (*ending*: negative, redemptive, survivor identity) design. The two *sexual violence* stories were child sexual abuse and adult sexual assault. The four *other types of trauma* stories were car accident, hurricane, life-threatening illness, or traumatic loss. Please refer to S1 File for copies of all vignettes.

In Study 1 (the mixed within- and between-subjects design), participants were assigned to all six trauma stories, in random order, with a randomized ending for each vignette. In Study 2 (the within-subjects design), participants were assigned to all six trauma stories, in random order, but were randomly assigned to read the same ending for all vignettes (e.g., to read all six trauma stories with a redemptive ending for each). Following each vignette to which they were randomly assigned, participants completed the same Likert-type questions assessing their perceptions of the trauma story and its storyteller. Following the story evaluations, participants completed additional self-report measures.

The study took participants approximately 30 minutes to complete. Participants from online samples were provided with an incentive of $4.00 if they passed the validity check question, completed the study in one sitting, and did not complete too quickly (in less than 15 minutes). University students were awarded course credit for their participation.

## Materials

**Trauma stories.**   In this 6 x 3 design, each of the 18 narratives describes a traumatic experience from the first-person perspective of an anonymous author. Narratives were written with a consistent level of detail, in a similar style, in a conversational tone, and at a fifth grade reading level (Flesch-Kincaid Grade Level = 5.1). Narratives were approximately 120 words each.

As stated previously, the six traumatic events were categorized as *sexual violence trauma* (child sexual abuse or adult sexual assault) versus *other types of traumas* (car accident, hurricane, life-threatening childhood illness, traumatic loss).

Regarding the less stigmatizing trauma stories (car accident, hurricane, life-threatening childhood illness, sudden traumatic loss of a loved one), we selected events representative of the main domains of (less stigmatizing) trauma as classified in the Diagnostic and Statistical Manual of Mental Disorders (DSM-5; [36]). Per the DSM-5, a traumatic event must involve "exposure to actual or threatened death, serious injury, or sexual violence" which is either experienced or witnessed [36]. We made the chronic life-threatening illness story a childhood (versus adulthood) story in order to match the chronic childhood sexual abuse story, to ensure that trauma type (more versus less stigmatizing) was not confounded with chronicity (acute versus chronic).

The three possible trauma story ending-types were *negative*, *redemptive*, or *survivor identity* ending. These unique endings manipulated whether the anonymous author was still negatively impacted by their trauma, if they had experienced personal redemption from the trauma, or if they had experienced personal redemption along with commitment to a mission to serve other victims as their life's work—a *survivor identity*.

Distinct negative, redemptive, and survivor identity endings were written for each of the 6 trauma stories in order to fit with the traumatic event that the narrator had experienced, yielding 18 trauma stories in all. Each ending was brief, limited several sentences. Please refer to OSF [31] and S1 File for all vignettes and endings.

**Participant perceptions of trauma storytellers.**   After reading each vignette, participants responded to evaluative questions about the story and its storyteller on a 5-point Likert-type scale from *strongly disagree* (1) to *strongly agree* (5). The difficulty sharing index included *n* = 8 items and assessed perceived difficulty sharing the traumatic event, (e.g., "If this were my

story, it would be emotionally difficult for me to share my story") with a Cronbach's alpha of .77 and .82 for Studies 1 and 2 respectively. The burden of sharing index included $n = 8$ items and measured perceived burdensomeness of sharing the story (e.g., "The author feels burdened to share this story with others") with a Cronbach's alpha of .72 and .82 respectively. The obligation to share index included $n = 20$ items and measured perceived obligation for the survivor to tell the story (e.g., "The author is obligated to share this story to make a difference in the lives of others"), this index had Cronbach's alpha of .94 and .96 respectively. Lastly, the likelihood of disclosure index had $n = 8$ items and measured perceived likelihood of disclosure (e.g., "The author is likely to share this story with others") with a Cronbach's alpha of .68 and .82 respectively. Items also assessed how likeable the storytellers were, with items such as "I would like to be friends with this author." For each domain (difficulty, burdensomeness, obligation, likelihood, likeable), the multiple items measuring that domain were averaged together into indices of each construct. For a full list of items and pre-registered scoring procedures, visit OSF [31].

**Participant self-reported trauma history.** Trauma history was indexed as the number of types of traumatic events that a participant self-reported to have experienced and/or witnessed. To create this index, all items on the 17-item Life Events Checklist (LEC-5) that participants selected *Happened to me* and/or *Witnessed it* were summed to yield a traumatic event history total score for each participant (range = 0–17; LEC-5; [37]). LEC-5 items represent events that are consistent with the definition of traumatic events in the DSM-5 [36].

## Analysis plan

All primary analyses were pre-registered on OSF [31]. An exploratory model, specified prior to analyzing the data but not pre-registered, is marked in the text (Exploratory Analyses 6–7). All analyses were conducted via ANOVAs in SPSS Version 26 [38]. All analyses were performed with the within-subjects dataset (Study 2) using factorial ANOVA. Hypotheses 4–5 were also tested with the mixed within- and between-subjects dataset (Study 1) using a multilevel model, or linear mixed effects model. With the exception of effect sizes, most descriptive and inferential statistics are presented in tables and not in the main text, to reduce redundancy. All analyses were performed with participant gender and personal trauma history specified as covariates, as pre-registered on OSF [31].

## Results

Hypotheses 1–3 examined differences in perceptions between story types (more versus less stigmatizing) with the same endings.

### H1 sexual trauma (versus other types of trauma) stories with negative endings

We hypothesized that sexual violence stories with negative endings would be perceived as *more difficult*, *more burdensome*, *less obligatory*, and *less likely to be shared* than other types of trauma stories with negative endings. This hypothesis was tested with a within-subjects design (Study 2). The hypothesis was supported for difficulty to share and for likelihood of sharing. Participants perceived sexual violence stories with negative endings as significantly *more difficult* to share ($\eta_p^2 = .08$) and *less likely* to be shared ($\eta_p^2 = .12$) compared to other types of trauma stories with negative endings. There were no significant differences for burden ($\eta_p^2 = .01$) or obligation ($\eta_p^2 = .01$). Results are presented in Table 2.

**Table 2. Hypothesis 1 means, standard deviations, and model results for perception of sharing by trauma and ending type.**

|  | Trauma Story Type | |  |  |  |
| --- | --- | --- | --- | --- | --- |
|  | Sexual Trauma | Other Trauma |  |  |  |
|  | M (SD) | M (SD) | F | p | np² |
| Negative Endings |  |  |  |  |  |
| Difficulty | 4.45 (.60) | 3.80 (.64) | 20.06 | < .001 | .081 |
| Burden | 3.18 (.95) | 2.79 (.67) | 2.61 | .107 | .011 |
| Obligation | 2.26 (.92) | 2.25 (.71) | 0.64 | .426 | .003 |
| Likely | 2.52 (.89) | 3.30 (.73) | 31.80 | < .001 | .122 |
| Redemptive Endings |  |  |  |  |  |
| Difficulty | 4.27 (.67) | 3.22 (.61) | 40.38 | < .001 | .148 |
| Burden | 2.79 (.86) | 2.42 (.63) | 1.97 | .162 | .008 |
| Obligation | 2.24 (.93) | 2.45 (.79) | 3.42 | .066 | .015 |
| Likely | 2.71 (.84) | 3.74 (.58) | 34.56 | < .001 | .130 |
| Survivor Identity Endings |  |  |  |  |  |
| Difficulty | 4.24 (.62) | 3.00 (.64) | 54.02 | < .001 | .190 |
| Burden | 2.71 (.84) | 2.31 (.62) | 9.10 | .003 | .038 |
| Obligation | 2.26 (.90) | 2.44 (.80) | 1.47 | .227 | .006 |
| Likely | 2.99 (.83) | 3.90 (.54) | 10.89 | .001 | .045 |

*Sexual Trauma* ($k$ = 2) refers to childhood sexual abuse or adult sexual assault. *Other Trauma* ($k$ = 4) refers to trauma caused naturally or by people accidentally (car accident, hurricane, childhood life-threatening illness), or traumatic loss (sudden death of a friend).

## H2 sexual trauma (versus other types of trauma) stories with redemptive endings

We hypothesized that sexual violence stories with redemptive endings will be perceived as *more difficult, more burdensome, more obligatory*, and *more likely to be shared* than other types of trauma stories with redemptive endings. This hypothesis was tested with a within-subjects design (Study 2). Results are presented in Table 2. The hypothesis was supported for difficulty to share. Participants perceived sexual violence stories with redemptive endings as significantly *more difficult* to share ($\eta_p^2$ = .15) than other types of trauma stories with redemptive endings. The direction of the effect for likelihood of sharing was the opposite of what we hypothesized; participants perceived sexual violence stories with redemptive endings as *less likely* to be shared ($\eta_p^2$ = .13) than other types of trauma stories with redemptive endings. There were no significant differences for burden ($\eta_p^2$ = .01) or obligation ($\eta_p^2$ = .02).

## H3 sexual trauma (versus other types of trauma) stories with survivor identity endings

We hypothesized that sexual violence stories with survivor identity endings will be perceived as *more difficult, more burdensome, more obligatory*, and *more likely to be shared* than other types of trauma stories with survivor identity endings. This hypothesis was tested with a within-subjects design (Study 2). Results are presented in Table 2. The hypothesis was supported for difficulty to share and for burden of sharing. Participants perceived sexual violence stories with survivor identity endings as significantly more difficult to share ($\eta_p^2$ = .19) and more burdensome to share ($\eta_p^2$ = .04) than other types of trauma stories with survivor identity endings. Consistent with the outcome of Hypothesis 2, the direction of the effect for perceived likelihood of sharing was the opposite of what we hypothesized; participants perceived sexual

violence stories with survivor identity endings as significantly *less likely* to be shared ($\eta_p^2 = .05$) than other types of trauma stories with survivor identity endings. There were no significant differences for obligation ($\eta_p^2 = .01$).

So far we have been comparing evaluations of more stigmatizing versus less stigmatizing trauma *story types*. We see that sexual trauma stories, regardless of ending, are perceived as more difficult to share, and less likely to share. We now turn to the results of Hypotheses 4–5, which examined differences in perceptions between *endings* (negative, redemptive, survivor identity), with story type (sexual trauma, other trauma) as a moderator.

### H4 trauma stories with negative versus positive endings and H5 moderation of effect of ending on perceptions by trauma type

We hypothesized that stories of traumatic events with *negative endings* for the storyteller would be perceived as *more difficult to share*, *more burdensome to share*, *less obligatory to share*, and *less likely to be shared* than stories of traumatic events with *positive endings* (redemptive followed by survivor identity). Further, we hypothesized that trauma type (sexual trauma versus other types of trauma) would moderate the proposed associations in Hypothesis 4. Hypotheses 4 and 5 were tested in both a mixed within- and between-subjects design (Study 1) and a within-subjects design (Study 2). Box plots for story ratings by ending and trauma type are presented in Figs 1 and 2.

**Study 1.** Hypothesis 4 was supported for difficulty to share, obligation to share, and likelihood of sharing. Participants perceived stories of traumatic events with negative endings for the storyteller as *more difficult*, *less obligatory*, and *less likely to be shared* than stories of traumatic events with redemptive endings, which in turn were perceived as *more difficult*, *less obligatory*, and *less likely to be shared* than stories of traumatic events with survivor identity endings. There was mixed support for Hypothesis 4 for perceived burden of sharing.

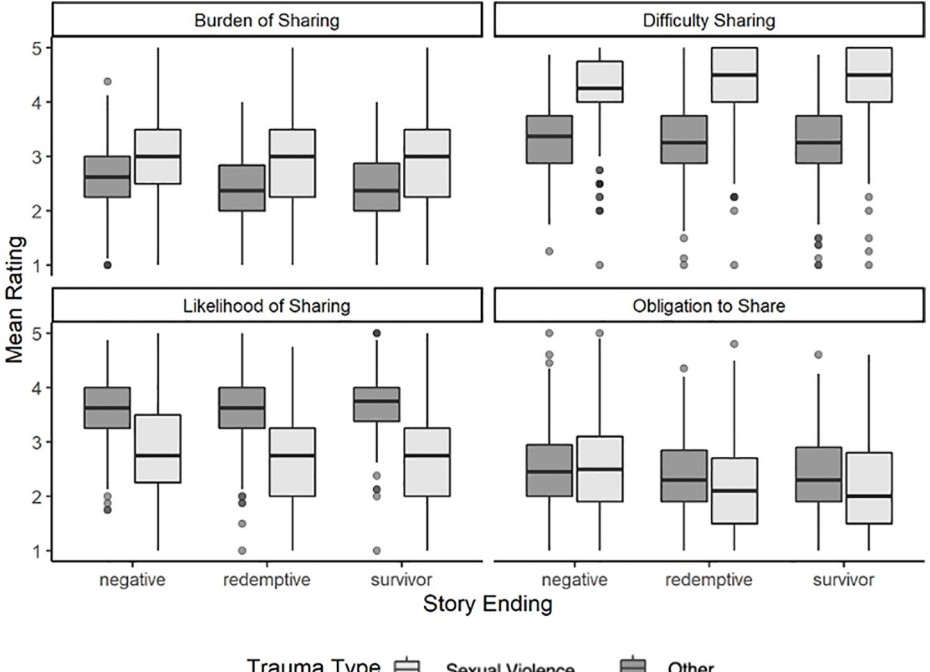

**Fig 1. Mean ratings of trauma stories by trauma type and ending (Study 1).**

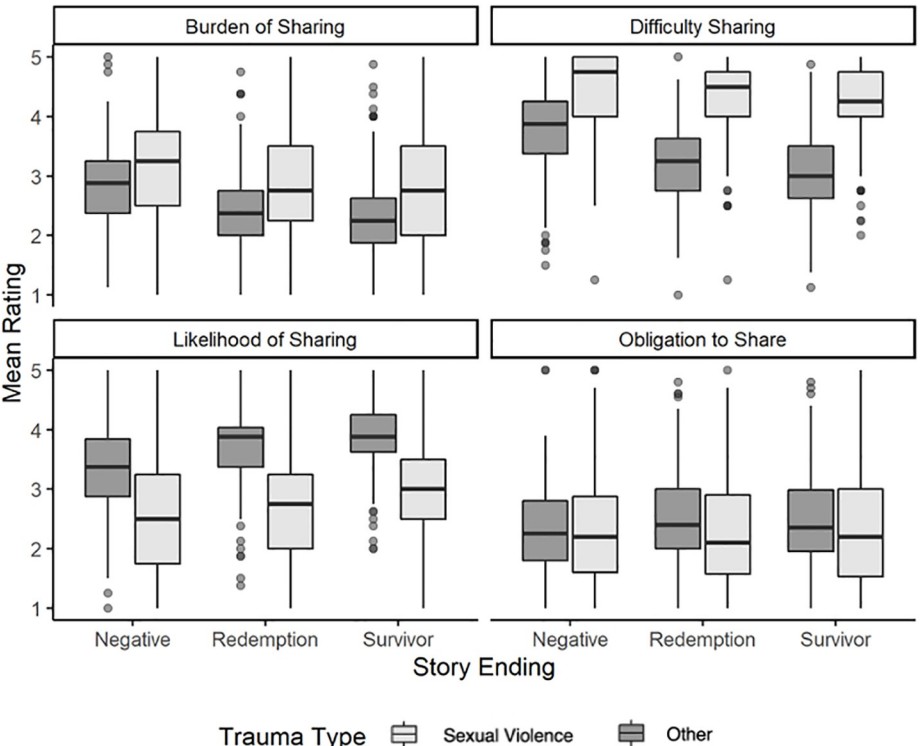

**Fig 2. Mean ratings of trauma stories by trauma type and ending (Study 2).**

Participants perceived stories of traumatic events with negative endings for the storyteller as more burdensome to share than stories of traumatic events with redemptive endings. However, stories of traumatic events with survivor (versus redemptive) endings were not perceived as any different in the perceived burden of sharing.

Hypothesis 5, the moderation hypothesis, was supported for difficulty to share, obligation to share, and likelihood of sharing, but not for perceived burden of sharing. Fig 3 presents the statistical results of the multi-level models.

In the case of perceived difficulty, less-stigmatizing trauma stories with redemptive endings were perceived as substantially easier to share than less-stigmatizing stories with negative endings. However, for sexual violence stories, having a redemptive ending did not make the story seem easier to share than if it had a negative ending. In the case of perceived obligation to share, trauma stories with redemptive (versus negative) endings increased the perceived obligation to share the story, but less so for sexual violence than for other types of traumas. Regarding likelihood of sharing, although positive endings (redemptive or survivor identity), on average, boost the perceived likelihood that the storyteller will share the story, moderation findings revealed that the impact of ending differs based on trauma type. Specifically, redemptive (versus negative) endings give less of a boost in sharing likelihood to sexual traumas than to other types of traumas, whereas survivor identity (versus redemptive) endings *increase* the likelihood of sharing for sexual traumas more than other types of traumas.

**Study 2.** Largely consistent with the results of Study 1, Hypothesis 4 was supported for difficulty to share and likelihood of sharing, with mixed support for perceived burden and no support for perceived obligation. Participants perceived stories of traumatic events with negative endings for the storyteller as *more difficult* to share than stories of traumatic events with

| Perceived Obligation | Estimate (SE) | t | p | LLCI | ULCI |
|---|---|---|---|---|---|
| Constant | 2.62 (0.09) | 29.58 | <.001 | 2.45 | 2.79 |
| Negative *vs.* Redemptive Ending | -0.43 (0.02) | -17.23 | <.001 | -0.47 | -0.38 |
| Survivor *vs.* Redemptive Ending | 0.14 (0.02) | 5.64 | <.001 | 0.09 | 0.19 |
| Sexual *vs.* Other Trauma | -0.20 (0.03) | -6.56 | <.001 | -0.25 | -0.14 |
| Negative*Sexual Trauma | 0.23 (0.04) | 5.35 | <.001 | 0.15 | 0.31 |
| Survivor*Sexual Trauma | 0.04 (0.04) | 1.01 | .312 | -0.04 | 0.13 |
| Covariates | Estimate (SE) | t | p | LLCI | ULCI |
| Participant Gender | -0.07 (0.04) | -1.46 | .145 | -0.15 | 0.02 |
| Participant Trauma History | -0.01 (0.01) | -0.87 | .387 | -0.03 | 0.01 |

| Perceived Difficulty | Estimate (SE) | t | p | LLCI | ULCI |
|---|---|---|---|---|---|
| Constant | 2.67 (0.07) | 39.72 | <.001 | 2.54 | 2.80 |
| Negative *vs.* Redemptive Ending | 0.72 (0.03) | 22.44 | <.001 | 0.66 | 0.78 |
| Survivor *vs.* Redemptive Ending | -0.17 (0.03) | -5.29 | <.001 | -0.23 | -0.11 |
| Sexual *vs.* Other Trauma | 1.12 (0.04) | 28.69 | <.001 | 1.04 | 1.20 |
| Negative*Sexual Trauma | -0.46 (0.06) | -8.21 | <.001 | -0.57 | -0.35 |
| Survivor*Sexual Trauma | 0.11 (0.06) | 1.94 | .052 | 0.00 | 0.22 |
| Covariates | Estimate (SE) | t | p | LLCI | ULCI |
| Participant Gender | 0.26 (0.03) | 7.85 | <.001 | 0.19 | 0.32 |
| Participant Trauma History | 0.003 (0.01) | 0.41 | .680 | -0.01 | 0.02 |

| Perceived Burden | Estimate (SE) | t | p | LLCI | ULCI |
|---|---|---|---|---|---|
| Constant | 2.41 (0.07) | 33.71 | <.001 | 2.27 | 2.55 |
| Negative *vs.* Redemptive Ending | 0.34 (0.03) | 10.99 | <.001 | 0.28 | 0.41 |
| Survivor *vs.* Redemptive Ending | -0.02 (0.03) | -0.58 | .563 | -0.08 | 0.04 |
| Sexual *vs.* Other Trauma | 0.47 (0.04) | 12.32 | <.001 | 0.40 | 0.55 |
| Negative*Sexual Trauma | -0.07 (0.05) | -1.37 | .172 | -0.18 | 0.03 |
| Survivor*Sexual Trauma | 0.06 (0.05) | 1.08 | .278 | -0.05 | 0.17 |
| Covariates | Estimate (SE) | t | p | LLCI | ULCI |
| Participant Gender | -0.03 (0.04) | -0.75 | .452 | -0.10 | 0.04 |
| Participant Trauma History | 0.002 (0.01) | 0.26 | .792 | -0.01 | 0.02 |

| Perceived Likelihood | Estimate (SE) | t | p | LLCI | ULCI |
|---|---|---|---|---|---|
| Constant | 3.73 (0.06) | 57.80 | <.001 | 3.60 | 3.86 |
| Negative *vs.* Redemptive Ending | -0.73 (0.03) | -25.60 | <.001 | -0.79 | -0.67 |
| Survivor *vs.* Redemptive Ending | 0.16 (0.03) | 5.55 | <.001 | 0.10 | 0.21 |
| Sexual *vs.* Other Trauma | -1.10 (0.03) | -31.80 | <.001 | -1.17 | -1.04 |
| Negative*Sexual Trauma | 0.28 (0.05) | 5.65 | <.001 | 0.18 | 0.38 |
| Survivor*Sexual Trauma | 0.25 (0.05) | 5.00 | <.001 | 0.15 | 0.35 |
| Covariates | Estimate (SE) | t | p | LLCI | ULCI |
| Participant Gender | 0.07 (0.03) | 2.35 | .019 | 0.01 | 0.14 |
| Participant Trauma History | 0.00 (0.01) | -0.31 | .756 | -0.01 | 0.01 |

**Fig 3. Hypotheses 4 and 5 multi-level model results for the effect of story ending on perceptions of sharing moderated by trauma type (Study 1).** Sexual vs. other trauma is a bivariate categorical variable coded as 1 if the story depicted childhood sexual abuse or adult sexual assault, and 0 if other type of trauma (car accident, hurricane, childhood life-threatening illness, or traumatic loss); *CI* = 95% confidence interval; *LL* = lower limit; *UL* = upper limit.

positive endings (redemptive or survivor identity), contrast estimate = .42 (SE = .04), *p* < .001, 95% CI [.34, .50]. Furthermore, participants perceived stories of traumatic events with redemptive endings as more difficult to share than stories with survivor identity endings (contrast estimate = .12 (SE = .05), *p* = .014, 95% CI [.02, .21].

Participants perceived stories of traumatic events with negative endings as *less likely* to be shared than stories of traumatic events with positive endings (redemptive or survivor identity), contrast estimate = -.43 (SE = .05), *p* < .001, 95% CI [-.52, -.33]. Furthermore, participants perceived stories of traumatic events with redemptive endings as less likely to be shared than stories with survivor identity endings (contrast estimate = -.24 (SE = .06), *p* < .001, 95% CI [-.35, -.13].

Participants perceived stories of traumatic events with negative endings as *more burdensome* to share than stories of traumatic events with positive endings (redemptive or survivor identity), contrast estimate = .41 (SE = .05), *p* < .001, 95% CI [-.06, .18]. However, there was no difference in the perceived burden of sharing between stories of traumatic events with redemptive versus survivor identity endings, contrast estimate = .06 (SE = .06), *p* = .318, 95% CI [-.06, .18].

Consistent with the results of Study 1, Hypothesis 5, the moderation hypothesis, was supported for difficulty to share, obligation to share, and likelihood of sharing, but not burden to share. Table 3 presents the results of the within-subjects factorial ANOVA models tested in Study 2.

**Table 3. Hypotheses 4 and 5 factorial ANOVA results for the effect of story ending on perceptions of sharing moderated by trauma type (Study 2).**

| | Trauma Story Type | | | | |
| | Sexual Trauma | Other Type of Trauma | | | |
| Perception of Sharing | M (SD) | M (SD) | F | p | $np^2$ |
|---|---|---|---|---|---|
| Difficulty | | | | | |
| Average All Endings | 4.30 (.65) | 3.34 (.71) | 41.39 | < .001 | .099 |
| Negative | 4.43 (.64) | 3.78 (.66) | | | |
| Redemptive | 4.26 (.67) | 3.23 (.61) | | | |
| Survivor | 4.22 (.63) | 3.03 (.65) | | | |
| Burden | | | | | |
| Average All Endings | 2.89 (.91) | 2.51 (.69) | .116 | .891 | .000 |
| Negative | 3.17 (.95) | 2.79 (.69) | | | |
| Redemptive | 2.78 (.86) | 2.42 (.63) | | | |
| Survivor | 2.73 (.85) | 2.33 (.65) | | | |
| Obligation | | | | | |
| Average All Endings | 2.27 (.93) | 2.39 (.79) | 8.94 | < .001 | .023 |
| Negative | 2.28 (.93) | 2.26 (.73) | | | |
| Redemptive | 2.26 (.93) | 2.46 (.79) | | | |
| Survivor | 2.29 (.94) | 2.46 (.82) | | | |
| Likely | | | | | |
| Average All Endings | 2.75 (.89) | 3.64 (.68) | 5.15 | .006 | .013 |
| Negative | 2.52 (.91) | 3.30 (.75) | | | |
| Redemptive | 2.71 (.85) | 3.73 (.60) | | | |
| Survivor | 3.02 (.83) | 3.90 (.55) | | | |

*Sexual Trauma* (k = 2) refers to childhood sexual abuse or adult sexual assault. *Other Trauma* (k = 4) refers to trauma caused naturally or by people accidentally (car accident, hurricane, childhood life-threatening illness), or traumatic loss (sudden death of a friend).

For perceived difficulty sharing, story endings had less impact on perceived difficulty for sexual traumas, than for other types of traumas. In other words, for less-stigmatizing traumas, redemptive and survivor identity endings resulted in a substantial *decrease* in perceived difficulty sharing, relative to negative endings. However, for sexual traumas, redemptive and survivor identity endings did not change the perceived difficulty of sharing.

Regarding perceived obligation to share, for less-stigmatizing traumas, redemptive and survivor identity endings (versus negative endings) substantially *increased* the perceived obligation to share the story. However, for sexual traumas, there was no effect of story ending (negative, redemptive, or survivor) on the perceived obligation to share the story.

Regarding perceived likelihood of sharing, redemptive (versus negative) endings gave *less* of a boost in sharing likelihood to sexual traumas than to other types of traumas. However, survivor identity (versus redemptive) endings *increased* the perceived likelihood of sharing for sexual traumas more than for other types of traumas.

## E6 Likeability of narrators of sexual (versus other) types of trauma stories with negative endings

This un-registered exploratory analysis was tested with a within-subjects design (Study 2). Consistent with our expectations, participants evaluated the narrators of sexual violence stories with negative endings (M = 2.82, SD = .50) as less likeable than the narrators of other types of trauma stories with negative endings (M = 2.92, SD = .45), F(1, 228), p = .002, $np^2$ = .04.

### E7 Likeability of narrators of sexual (versus other) types of trauma stories with positive endings

This un-registered exploratory analysis was tested with a within-subjects design (Study 2). Consistent with our expectations, participants evaluated the narrators of sexual violence stories with redemptive endings ($M$ = 3.33, $SD$ = .53) as less likeable than the narrators of other types of trauma stories with redemptive endings ($M$ = 3.53, $SD$ = .41), $F(1, 232)$ = 11.64, $p$ = .001, $np^2$ = .05. Participants also evaluated the narrators of sexual violence stories with survivor identity endings ($M$ = 3.36, $SD$ = .50) as less likeable than the narrators of other types of trauma stories with survivor identity endings ($M$ = 3.55, $SD$ = .43), $F(1, 230)$ = 7.54, $p$ = .007, $np^2$ = .03.

## Discussion

Most individuals who have experienced sexual assault and other forms of interpersonal violence do not disclose their experiences to others, in part due to the stigma attached to such experiences. Both delayed disclosure and negative social reactions to disclosure (e.g., victim-blaming) can exacerbate posttraumatic stress and disrupt help-seeking [39,40]. The tendency not to share traumatic experiences can also create a negative feedback loop where these silenced experiences seem uncommon or taboo to discuss. Although the Me Too movement has encouraged survivors of sexual violence to come forward and tell their stories in unprecedented numbers, surprisingly little is known about how contemporary audiences perceive and interpret trauma stories and their storytellers.

In this vignette-based experimental study, 1,872 U.S. adults were randomly assigned to experimental conditions, or trauma stories that varied based on the type of trauma and the ending of the story. We based our hypotheses about story-telling perceptions on theory and empirical work bridging the fields of clinical, trauma, and narrative psychology. Across a set of five pre-registered hypotheses and two exploratory analyses, we proposed that public perceptions of trauma stories in the U.S. would be informed by cultural values and assumptions about why bad things happen to people and about how stories about these experiences should be told.

Overall, our study findings provide robust evidence—across a broad range of participants, analytic approaches, both within- and between-subjects—that participants perceived sexual trauma stories as more difficult to tell and less likely to be told than other, less stigmatizing trauma stories, even when stories end positively. Participants in this study also perceived narrators of sexual violence stories as less likeable than narrators of other types of trauma stories, even when stories ended on a positive note. Positive endings (a redemptive ending or a special type of redemption that we term *survivor identity*), in other words, are perceived less favorably in the case of sexual trauma, in ways that are unpacked below.

### Supported hypotheses

In this study, participants agreed that sexual violence stories would be difficult to tell, but were more neutral as to whether other types of trauma stories would be difficult to disclose, with large effect sizes. However, in a robust finding replicated across samples and analytic approaches, having a positive story ending did *not* make sexual violence stories seem any easier to tell or more likely to be told—only less-stigmatizing stories got this advantage. Whether endings were positive or negative, sexual violence stories were perceived as hard to tell and unlikely to be told. Similarly, even when story endings were positive, participants judged narrators of sexual violence stories as less likeable than narrators of other types of trauma stories (this was an exploratory analysis). Part of this pattern of findings was contrary to our

hypotheses: we predicted that having positive endings would make sexual violence stories seem *more likely* to be told, while still being perceived as more difficult to tell.

These findings portray a chilly climate for sharing stories of sexual violence in the U.S. On the one hand, this is to be expected, given low disclosure rates. In a novel contribution, these findings show that even when a person who has experienced a highly stigmatizing form of trauma demonstrates resilient coping (healing has occurred, emotions no longer raw, day-to-day functioning restored), the public has unfavorable perceptions of storyteller and the story-telling process. The implications of this are marked, as they suggest that a socially desirable narrative structure of positive transformation—redemption—does not extend to survivors who are most in need of a favorable response to their stories. There is something about the experience of sexual violence itself, then, that remains a "mark of failure or shame," no matter what meaning can be made of it [10 p. 2].

For survivors tasked with telling their stories of sexual violence to audiences large and small, there are painful and practical implications here. At a fundamental level, the development of a sense of self depends, in part, on being able to tell a personal narrative and on having that narrative recognized by others. Personal storytelling helps people understand themselves and their lives, and lets them be truly known to others [41,23].When public perceptions reflect the assumption that sexual violence survivors are not able to tell their stories and that it would be difficult or burdensome to do so, survivors lose out on an opportunity to be known. For instance, survivors may want to be recognized for the depths of their continued suffering, or to be celebrated for being able to rise up and make coherent meaning (and a meaningful contribution) out of the unspeakable. Furthermore, when someone does tell a story but receives a message not that what happened to them is bad, but that *they* are bad (or at the very least, less likeable), shame and self-blame can be expected. To carry a "mark of failure or shame," to conceal socially undesirable aspects of one's experience, may distance the self from others and tarnish the survivor's identity.

If audiences feel uncomfortable with sexual violence stories, even when storytellers are survivor-advocates who embody culturally valued traits, the public may be less willing to sustain the years of attention and action needed to hold authorities accountable for stopping sexual abuse and preventing abuse from happening in the future. Collective indifference perpetuates the institutional inaction that enables abuses to persist, as in the case of the USA Gymnastics organization and the decades-long sexual abuse of hundreds of child athletes by team doctor Larry Nassar [42,43,44]. On the one hand, the Me Too movement has opened more public airtime for #MeToo stories and survivor-advocate voices in the past several years. Founded in 2006 by activist Tarana Burke, the Me Too movement was further popularized in 2016 by survivor-advocates in the entertainment industry, in the wake of rape and sexual assault allegations against Hollywood producer Harvey Weinstein. The recent surge in #MeToo stories has re-animated political consciousness of sexual violence and created opportunities for the public to demand accountability from some high-profile perpetrators in the entertainment industry. However, despite more public airtime for #MeToo in the past several years, rates of sexual violence have increased over the same time period in the U.S. According to national crime statistics published in 2019, the percentage of U.S. residents age 12 or older who were victims of sexual violence doubled from 2015 to 2018 (from 1.4 to 2.7 per 1,000 persons; [45]. The rate of violent victimizations reported to police did not change in that time, while the rate *not* reported increased, meaning the higher rates of sexual violence over the past few years cannot be explained by an uptick of reported events than would typically go unreported. The causes of sexual violence prevalence rates are, of course, complex and multi-determined. At the very least, it can be stated that the Me Too movement has not coincided with a decrease in sexual violence victimization rates in the U.S. There may remain, as some have argued, a stubborn,

pervasive societal "unawareness and illiteracy" around sexual violence [46 p.1], including its frequency and common victim-perpetrator dynamics.

## Unsupported hypotheses

Regarding the perceived obligation to tell trauma stories, we proposed that there would be a paradoxical set of expectations for sexual trauma survivors: to be silent about their experiences *and* to tell others about their experiences, if they can tell a story with a happy, and empowered, ending. Contrary to our expectations, positive endings did *not* increase the perceived obligation to tell sexual violence stories. Positive endings only boosted the perceived obligation to share other, less stigmatizing types of trauma stories. Given U.S. popular culture's celebration of personal responsibility and of redemptive stories of overcoming adversity, we anticipated that there would be a perceived obligation to share stories of overcoming sexual traumas. Sharing such stories can inspire others and demonstrate that personal resilience is possible even after the worst that humans can do to one another. However, our results suggest that the U.S. celebration of redemption does not supersede the cultural stigma attached to sexual violence. That is, audiences may perceive there to be social devaluation and loss of status among those who have experienced sexual violence [47], *even if the person has transcended their trauma.*

Another interpretation of this finding is that audiences recognize the chilly climate for sexual violence stories and, from a place of compassion, do not believe these survivors should be obligated to tell their stories. This compassion would be well-placed, for at least two reasons. One, we would not want to obligate anyone to share a personal story, let alone coerce them, an echo of the loss of agency and free choice faced by victims of violence. Two, we would not want to obligate survivors of sexual violence to tell their stories publicly in the absence of justice being available to them. As an example, although many survivors of child sexual abuse wait until adulthood to tell anyone—the average age of disclosure is 52—many U.S. states have short statutes of limitation that prevent adult survivors from filing civil or criminal lawsuits against perpetrators [48].

Regarding the perceived burden of sharing a trauma story, support for our hypotheses was mixed. In general, as expected, participants agreed that sexual trauma (versus other types of trauma) stories and stories with negative (versus positive) endings would be more burdensome to tell. However, our hypothesis that positive endings would increase the perceived burden to share sexual traumas was supported only for survivor identity endings. That is, participants perceived sexual violence stories with survivor identity endings as slightly *more burdensome* to share than other types of trauma stories with survivor identity endings. This finding should be interpreted with caution because, in general, participants leaned toward disagreeing that *any* trauma story was burdensome to tell—except for sexual violence stories with negative endings. Put another way, although participants disagreed that sexual *and* other trauma stories with survivor identity endings would be burdensome to tell, they disagreed a little bit less for sexual traumas.

## Clinical implications

Our findings reflect ambivalence surrounding sexual violence in the U.S., with direct implications for survivor mental health. On the one hand, people seem to recognize that it is more difficult and less likely to tell sexual violence stories than other types of trauma stories. Moreover, advocates and scholars have drawn attention to systemic and societal barriers to sexual violence story-telling for highly stigmatized, marginalized, and potentially isolated survivors, such as people with disabilities [49] and immigrants (who may or may not have limited-English proficiency, and be undocumented or part of a "mixed status" immigrant family; [50]). And

yet, sexual assault victims are questioned as to why they do not say something sooner, or at all [51]. Some survivors have pushed back publicly against this hurtful, dismissive perception (for e.g., see the hashtag #WhyIDidntReport), but others internalize the blame, compounding their suffering.

There are many professionals who bear witness to adolescents and adults telling stories of sexual violence in an official or public way: therapists, school counselors, teachers, healthcare providers, law enforcement, journalists, and human resources personnel. These professionals can be encouraged to recognize the enormous difficulty of sharing a story of sexual violence and to offer the storyteller validation around this difficulty. Sexual violence survivors can be affirmed for their essential worth and value, no matter what degrading events they have endured or how triumphant their recovery.

Industrial and organizational psychologists must also consider their role in helping to create safe workplaces, in light of how difficult and unlikely it is (and is perceived to be) to tell a story of sexual violence. The challenges that survivors face in sharing sexual violence stories, even when they have "positive endings," benefit perpetrators, especially the serial offenders who have been the first to fall in the Me Too movement. How can the policies of organizations be shaped to incentivize the authentic, timely story-telling of survivors, to adequately protect these storytellers from retaliation, and to afford the swift pursuit of justice on their behalf?

### Limitations and future directions

The findings of this vignette-based experimental study must be considered in light of several limitations. First, while we recruited nationally for this online study and randomly assigned participants to experimental conditions, in three of the four data collection sites, female-identified adults were over-represented, and in all data collection sites, adults who identified as White were over-represented. Although all estimates of effects in our analyses accounted for participant gender, future research must unpack the degree to which findings generalize across genders and other dimensions of identity, including race, ethnic identity, and degree of U.S. acculturation. Participant trauma history, a covariate in the analysis, was measured in broad strokes (as total count of lifetime traumas), as we did not specify a priori hypotheses about how an individual's own more stigmatizing (versus less-stigmatizing) trauma history and disclosure experiences, if any, would predict story-telling perceptions.

Future research can consider the complex question of how an individual's gender identity and trauma history type interact to differentially predict story perceptions. Addressing this question is complicated by the common co-occurrence of both interpersonal violence and other types of traumas [52], and by the confounding of gender and trauma type (males experience a higher rate of traumas perpetrated by strangers, whereas females and transgender men and women experience higher rates of traumas perpetrated in relationships; [53,54]). Difficult to parse as a trauma-history-by-gender interaction may be, the effort would help illuminate the conditions under which individuals in the U.S. accommodate to versus resist the stigmatization of sexual violence survivors and the dominant cultural value of redemption [6].

We designed this study based on theory on the cultural stigma of sexual violence which foregrounds the role of stigma as it applies to the act of violence itself. While the theory-driven nature of this study is a strength, future research must grapple with additional social and cultural contexts that accentuate the stigma of sexual violence. These layers of stigma likely have implications for perceptions of and reactions to trauma storytellers. For instance, male-identified survivors of sexual violence arguably experience more (at the very least, different) stigma than female-identified survivors [55]; interpersonal violence in queer relationships, especially for lesbian and bisexual survivors, carries additional stigma [56]; and people of color may

experience additional blame and stigma related to their experiences of violence [57]. And of course, there are likely important differences in cultural stigma and preference for redemption between and within U.S. regions, states, and communities that remain to be explored.

Sexual violence stories are representative of the stories being shared in the international Me Too movement, which will be familiar to many readers. Importantly, there may be unexplored differences in story-telling perceptions across the major types of interpersonal violence (sexual, physical, psychological). For instance, stories of psychological violence (such as chronically humiliating, controlling, bullying treatment in a relationship) may be perceived as even more difficult to share and their storytellers least likeable, relative to acts of sexual and physical assault that leave visible bruises or are brought to greater public light in a criminal court of law. Likewise, as a more discreditable form of interpersonal violence, psychological violence may be perceived as even more difficult to narratively "redeem" than sexual or physical violence. Future experimental studies can examine whether perceptions of interpersonal violence storytelling differ based on the form of interpersonal violence (sexual, physical, psychological), *and* on the trauma's visibility and concealability in the aftermath of violence [10].

In sum, the experience of sexual violence is a culturally shaped phenomenon, and both survivors and those bearing witness to trauma have a range of intersecting identities that will inform their reactions to the event. This study provides a robust set of findings that stand in their own right, while inviting others to contribute to a tapestry of future research on the trauma story-telling process across diverse samples and methodologies.

## Conclusions

In the U.S., there is a strong cultural preference for redemptive stories of overcoming adversity. Audiences admire and celebrate redemptive stories and their storytellers. But our novel findings suggest that the type of adversity matters. Audiences perceive sexual violence stories and their storytellers less favorably than other, less stigmatizing traumas like natural disaster, even when the story ends with redemption. The ramifications of this cultural preference need attention. To effectively support sexual violence survivors, and to intervene on institutional and societal forces that enable abuse to continue, the public must bear witness to stories that many wish did not exist.

## Supporting information

**S1 File. Trauma stories.** Trauma story vignettes and endings to which participants could be randomly assigned.
(PDF)

## Acknowledgments

We wish to thank the associates of the Center for Cross-Cultural Research at Western Washington University for their valuable conversation and feedback during the design phases of this project.

## Author Contributions

**Conceptualization:** Brianna C. Delker, Kate C. McLean.

**Formal analysis:** Brianna C. Delker.

**Funding acquisition:** Brianna C. Delker, Kate C. McLean.

**Investigation:** Rowan Salton.

**Methodology:** Brianna C. Delker, Kate C. McLean.

**Project administration:** Brianna C. Delker, Kate C. McLean.

**Resources:** Brianna C. Delker, Kate C. McLean, Moin Syed.

**Software:** Moin Syed.

**Supervision:** Brianna C. Delker, Kate C. McLean.

**Validation:** Moin Syed.

**Visualization:** Moin Syed.

**Writing – original draft:** Brianna C. Delker.

**Writing – review & editing:** Rowan Salton, Kate C. McLean, Moin Syed.

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
