## [Decision Letter · Decision Letter 0]

13 Mar 2020

PONE-D-20-02734

Who has to tell their trauma story and how hard will it be? Influence of cultural stigma and narrative redemption on the storying of interpersonal violence

PLOS ONE

Dear Dr. Delker,

Thank you for submitting your manuscript to PLOS ONE. After careful consideration, we feel that it has merit but does not fully meet PLOS ONE’s publication criteria as it currently stands. Therefore, we invite you to submit a revised version of the manuscript that addresses the points raised during the review process.

The authors present a generally well written paper that is highly timely, relevant and useful to our broader knowledge of perceptions of traumatic experience stories. I agree with reviewer 1 that the paper does not yet meet PLOS ONE publication criteria surrounding sufficient detail and intelligibility, but can meet those criteria through revision. My own specific comments to inform your revision are included below, complementing the feedback also provided by Reviewer 1.

We would appreciate receiving your revised manuscript by Apr 27 2020 11:59PM. To enhance the reproducibility of your results, we recommend that if applicable you deposit your laboratory protocols in protocols.io, where a protocol can be assigned its own identifier (DOI) such that it can be cited independently in the future. For instructions see: http://journals.plos.org/plosone/s/submission-guidelines#loc-laboratory-protocols

We look forward to receiving your revised manuscript.

Kind regards,

Whitney S. Rice, DrPH

Academic Editor

PLOS ONE

Additional Editor Comments (if provided):

Overall

• The authors mention the hashtag #MeToo in several places but do not describe what they are referring to. Many readers may be familiar with this movement, but others may not. Please describe/define what #MeToo is at some point in the text, preferably near first mention.

• Consider removing the use of the "non-stigmatizing" term to refer to any trauma. You could simply refer to the traumas as sexual assault vs. other trauma, or more vs. less stigmatizing, or other language. In certain circumstances and perhaps for certain people, each of the events in this category of trauma could be subject to cultural stigma (see Kaushansky et al 2016 Chronic Illness, Pitman et al 2016 Journal of Psychosomatic Research, Krzemieniecki and Gabriel 2019 Journal of Mental Health [which provides support counter to the hypothesis that sexual assault stigma is greater than car accident stigma wording], etc.).

• The authors justify in multiple places the hypothesis around degree of stigma associated with sexual assault vs. the other traumatic events. Have you considered that part of the reason for distinction, in addition to that named in the present draft, could be surrounding sex and sexuality are taboo and stigmatized in many cultures, and people have difficulty discussing sex period let alone sexual assault?

The Present Study

• Page 11, Line 269: The authors are missing the one of the brackets in the parenthesis here.

Methods

• Participants section: Do the authors have more information about the geographic distribution of the study sample across US states or regions? This information could provide readers with important context about the study, considering that cultural stigma may differ from region to region and state to state.

Discussion

• Page 30, Line 667: The use of "distorted thinking" reads like a value judgment, and doesn't acknowledge that the findings may reflect more nuance. Consider revised use of language and tone here.

• Page 31, Line 669: See above comment - "by way of collective mental gymnastics" also reads similarly. Consider removing this piece of the sentence.

• Page 31, Line 682: Explain for the reader who isn't reading between the lines - what do you mean by "It is to the benefit of perpetrators"? What is "It" and how does that thing/tendency benefit perpetrators?

Journal Requirements:

Reviewers' comments:

Reviewer's Responses to Questions

**Comments to the Author**

1. Is the manuscript technically sound, and do the data support the conclusions?

Reviewer #1: Yes

Reviewer #2: Yes

2. Has the statistical analysis been performed appropriately and rigorously? 

Reviewer #1: Yes

Reviewer #2: Yes

3. Have the authors made all data underlying the findings in their manuscript fully available?

Reviewer #1: Yes

Reviewer #2: Yes

4. Is the manuscript presented in an intelligible fashion and written in standard English?

Reviewer #1: Yes

Reviewer #2: Yes

5. Review Comments to the Author

Reviewer #1: Overall, the topic of this study is very relevant and timely. There are considerable strengths, and the authors have done a great job in trying to provide a detailed account of their research. Nevertheless, there are a few areas that still need to be looked at to make this paper stronger. One of the biggest areas of improvement in the manuscript is a choice of words. I encourage the authors to carefully review the manuscript and adhere to scientific/formal writing. Here are my review comments:

Abstract

• It would be great to make sure that it is clear, early on, whose perceptions are shaped by the cultural stigma surrounding interpersonal violence and a cultural preference for positive endings to adversity. The authors clarified this later in the paper, but it is good to do that early on.

• If possible, avoid the use of the term redemptive or redemption throughout the manuscript. It sets off some readers as the term is very loaded and contrary to the point you are making in their paper. Redemption is the act of or process of redeeming, and it externalizes the drivers of a positive ending to adversity (i.e., it diminishes the role, bravery, and courage of the victims in overcoming the adversity). I suggest using a “positive ending” instead.

• The sentence on line 87 “no more obligatory to tell” may need revision unless there was a time when it was obligatory to tell.

Background/introduction

• Beginning of line 93, “I” should be a small case

• Please define the following words: interpersonal violence, cultural stigma, and cultural stigma.

• It is not clear why you wanted to use the broader term, interpersonal violence, through the paper while you, in fact, looked only at child sexual abuse and adult sexual assault.

• Please Consider revising some extra-long sentences through the paper. It is affecting the clarity of your message. Example, line 100-102, 104-107. There are several places throughout the manuscript where long sentences made the sections not only hard to read, but the key messages are lost in the weed.

• I encourage you to avoid the use of certain words such as “Surprisingly,” Distressingly, “Absurd,” “Not surprisingly,” “Understandably,” “chilly climate,” etc. These words could through your readers off because they try to impose your personal views on the issues on your readers.

• I suggest removing the word “Act of God” and use “natural disaster” for the same reason I mention in my preceding comments.

• Please clarify how cultural stigma offset cultural preference.

• Please also clarify how media, the internet, etc. are cultural products or define what you meant by cultural products.

• I would change the word “degradation” - line 208.

• The sentence on line 217-218 “ The perception that survivors of interpersonal violence must bear the burden to tell their stories is consistent with the implicit cultural assumption that victims are personally responsible for what happens to them,” contradict with eh what is stated on line 161-162 regarding the invisible nature of the interpersonal violence.

Method

• Something is off on line 287 regarding average age and age range

• Please review your data in Table 1. There several issues with the percentage values.

• Your participants were self-selected. You may want to explain the impact of self-selection bias on your finding somewhere in your manuscript, preferably in the limitation section.

• Lined 297: I would use “incentive” instead of “compensation” unless you fully compensated the participant based on the locally approved hourly wage.

• Do your participants, especially university students, know each other? How that defies the purpose of the random assignment? What measures were taken to reduce the risks? How might that have affected your results?

Discussion/Conclusion

• It appears that you made a very strong generalization. I do not think you nationally representative data to make an inference to the general US population.

• There are situations where individuals are legally obligated to tell the act of violence. Your sentence on line 646 may need to reflect circumstances where individuals have a legal obligation to tell the story.

• I do not think your findings reflect that the US society has deep ambivalence and distorted thinking about interpersonal violence. This is really making a robust inference to the general population which I do not believe you have enough statistical power to make that generalization.

• You suggest that professionals can be encouraged to recognize the difficulty of sharing a story of interpersonal violence and offer validation around this difficulty, but you did not offer action steps. Also, since most of the victimizations happen in the community among the margin lied group (e.g., domestic workers), what other recommendations you give to those working to help the most vulnerable group?

Reviewer #2: This was an excellent paper, congratulations. It was very well written, intellectually stimulating, conceptually mature, and highly significant. The impact for the field is huge. I might also recommend reading this article: https://medium.com/@p.sawrikar/hypocritical-wiring-and-its-limits-on-empathy-the-sense-of-agency-bias-9b9aeaaba3b3?source=friends_link&sk=774be8c483b91140dadedf988f889792

6. PLOS authors have the option to publish the peer review history of their article (what does this mean?). If published, this will include your full peer review and any attached files.

Reviewer #1: No

Reviewer #2: No

---

## [Author Response · Author response to Decision Letter 0]

28 Apr 2020

April 27, 2020

Response to Reviewers

Empirical Article Manuscript PONE-D-20-02734

“Who has to tell their trauma story and how hard will it be? Influence of cultural stigma and narrative redemption on the storying of sexual violence”

Additional Editor Comments (if provided):

Overall

• The authors mention the hashtag #MeToo in several places but do not describe what they are referring to. Many readers may be familiar with this movement, but others may not. Please describe/define what #MeToo is at some point in the text, preferably near first mention.

>>>Thank you for pointing this out. We have added a definition of #MeToo near the first mention, in the middle of the first paragraph of the Introduction.

• Consider removing the use of the "non-stigmatizing" term to refer to any trauma. You could simply refer to the traumas as sexual assault vs. other trauma, or more vs. less stigmatizing, or other language. In certain circumstances and perhaps for certain people, each of the events in this category of trauma could be subject to cultural stigma (see Kaushansky et al 2016 Chronic Illness, Pitman et al 2016 Journal of Psychosomatic Research, Krzemieniecki and Gabriel 2019 Journal of Mental Health [which provides support counter to the hypothesis that sexual assault stigma is greater than car accident stigma wording], etc.).

>>>Thank you for your feedback about the reality that in certain circumstances (and for certain clinical or demographic populations), any of the traumatic events could be subject to cultural stigma. We agree with this point and had been grappling with the using of “non-stigmatizing” from the outset of the project. We have looked at the articles you mentioned and added two of them as citations in the manuscript. 

More to the point, we have removed the use of the “non-stigmatizing” term to refer to any trauma. Instead, we use the terms “more stigmatizing” versus “less stigmatizing” when we introduce the conceptual definition of these types of trauma (p. 5). For the remainder of the manuscript, we favor the terms “sexual violence versus other types of trauma” instead of “more versus less stigmatizing traumas.” We agree this is the most accurate language, as our operationalizations of “more stigmatizing traumas” were both forms of sexual violence. We also changed the manuscript title from “…storying of interpersonal violence” to “…storying of sexual violence.” In the Discussion section, we present suggestions for future research that considers the degree to which audience reactions to sexual violence stories generalize to other types of interpersonal violence stories.

• The authors justify in multiple places the hypothesis around degree of stigma associated with sexual assault vs. the other traumatic events. Have you considered that part of the reason for distinction, in addition to that named in the present draft, could be surrounding sex and sexuality are taboo and stigmatized in many cultures, and people have difficulty discussing sex period let alone sexual assault?

>>>Thank you for this note. We did not measure general stigma toward sex/sexuality in this study, but we added an explicit acknowledgement about stigmatization of sex/sexuality in many cultures, in the introduction.

The Present Study

• Page 11, Line 269: The authors are missing the one of the brackets in the parenthesis here.

>>>Thank you – the brackets appear in our Microsoft Word version. I am not sure what happened in the format change to PDF.

Methods

• Participants section: Do the authors have more information about the geographic distribution of the study sample across US states or regions? This information could provide readers with important context about the study, considering that cultural stigma may differ from region to region and state to state.

>>>To increase anonymity and for confidentiality reasons, we did not collect location data from our national sample participants recruited via MTurk and Qualtrics Panels. MTurk and Qualtrics Panels participants could be from anywhere in the country. However, we agree with the implication that there may be important differences in stigma toward trauma stories and cultural preference for redemption both within and between regions and states in the U.S. We have added this point to the Discussion section of the manuscript.

Discussion

• Page 30, Line 667: The use of "distorted thinking" reads like a value judgment, and doesn't acknowledge that the findings may reflect more nuance. Consider revised use of language and tone here.

>>>We removed the term ‘distorted thinking’ and made a read-through of the Discussion to edit tone.

• Page 31, Line 669: See above comment - "by way of collective mental gymnastics" also reads similarly. Consider removing this piece of the sentence.

>>>Thank you, we have removed this piece of the sentence.

• Page 31, Line 682: Explain for the reader who isn't reading between the lines - what do you mean by "It is to the benefit of perpetrators"? What is "It" and how does that thing/tendency benefit perpetrators?

>>>We have edited this sentence to clarify that ‘it’ refers to survivor difficulties sharing stories of sexual violence.

Journal Requirements:

>>>Thank you. After reviewing the style templates above and re-reviewing the PLOS ONE style requirements, we have made the following style changes to the manuscript:

>Formatted the Title Page according to PLOS ONE style requirements (e.g., sentence case for title, author affiliations).

>Reformatted the manuscript according to PLOS ONE style requirements (e.g., heading and subheading font styles, sentence case for headings, cite figures as “Fig 1,” numeric reference list, no funding info in Acknowledgements, etc.)

>Renamed Figure 1a and 1b “Figs 1 and 2” and added the titles and captions below their first mention in text, and changed file format of figures to .tiff.

>>>We have added a Supporting Information Caption to the Ending Section, reformatted the supporting information in text to “S1 File,” and renamed supporting information file to “S1_File.pdf.”

Reviewers' comments:

Reviewer's Responses to Questions

Comments to the Author

1. Is the manuscript technically sound, and do the data support the conclusions?

Reviewer #1: Yes

Reviewer #2: Yes

2. Has the statistical analysis been performed appropriately and rigorously?

Reviewer #1: Yes

Reviewer #2: Yes

3. Have the authors made all data underlying the findings in their manuscript fully available?

Reviewer #1: Yes

Reviewer #2: Yes

4. Is the manuscript presented in an intelligible fashion and written in standard English?

Reviewer #1: Yes

Reviewer #2: Yes

5. Review Comments to the Author

Reviewer #1: 

Overall, the topic of this study is very relevant and timely. There are considerable strengths, and the authors have done a great job in trying to provide a detailed account of their research. 

>>>Thank you for these favorable comments about our manuscript.

Nevertheless, there are a few areas that still need to be looked at to make this paper stronger. One of the biggest areas of improvement in the manuscript is a choice of words. I encourage the authors to carefully review the manuscript and adhere to scientific/formal writing. Here are my review comments:

Abstract

• It would be great to make sure that it is clear, early on, whose perceptions are shaped by the cultural stigma surrounding interpersonal violence and a cultural preference for positive endings to adversity. The authors clarified this later in the paper, but it is good to do that early on.

>>The Abstract of the paper states that the perceptions in question refer to adult audiences in the United States (U.S.) and that the study participants consist of four samples of U.S. adults. The first paragraph of the introduction refers to “general” and “mainstream” audiences of U.S. adults as being the focus of this investigation, and the second page of the introduction clarifies what we mean by U.S. “culture” and mainstream values. If readers would benefit from greater clarity, we would be happy to make adjustments, but currently we are not perceiving the ambiguity that this reviewer is noting.

• If possible, avoid the use of the term redemptive or redemption throughout the manuscript. It sets off some readers as the term is very loaded and contrary to the point you are making in their paper. Redemption is the act of or process of redeeming, and it externalizes the drivers of a positive ending to adversity (i.e., it diminishes the role, bravery, and courage of the victims in overcoming the adversity). I suggest using a “positive ending” instead.

>>>Respectfully, we are puzzled by this suggestion to avoid use of the terms redemptive or redemption throughout the manuscript. We designed this study in order to investigate the influence of redemptive story-telling on perceptions of sexual (versus other) traumas. Although redemptive endings are indeed positive, the meaning of “redemption” extends beyond positivity and is historically grounded in U.S. cultural history. Redemption from trauma implies that victims have transcended or triumphed over adversity and personally grown from their experiences; as such, it certainly does not diminish the courage of victims (if anything, it celebrates it). We regret if any reader perceived a tone of diminishment. The cultural preference for redemptive stories among U.S. audiences is theoretically grounded and empirically supported with strong evidence (e.g., McLean, Delker, Salton, Dunlop, & Syed, 2020). As such, we use this term because it accurately describes a phenomenon we are trying to understand. 

• The sentence on line 87 “no more obligatory to tell” may need revision unless there was a time when it was obligatory to tell.

>>>Thank you, we have clarified this sentence.

Background/introduction

• Beginning of line 93, “I” should be a small case

>>In the PDF version of the manuscript draft generated by the editorial manager system, the beginning of line 93 is the first line of the manuscript. The only capitalized “I” in this sentence is “In,” the first word of the manuscript, which needs to be capitalized.

• Please define the following words: interpersonal violence, cultural stigma, and cultural stigma.

>>>We define “interpersonal violence” on p. 2 as trauma caused by people intentionally. Cultural stigma is defined in the subsection of the introduction “Cultural Stigma Surrounding the Experience of Sexual Violence Informs Reactions to Disclosure” (pp. 4-5).

• It is not clear why you wanted to use the broader term, interpersonal violence, through the paper while you, in fact, looked only at child sexual abuse and adult sexual assault.

>>>This call for greater precision is appreciated and, in response to this feedback and the Editor’s feedback, we have edited the manuscript to use the more specific term, “sexual violence,” throughout the manuscript (instead of the broader term “interpersonal violence”).

• Please Consider revising some extra-long sentences through the paper. It is affecting the clarity of your message. Example, line 100-102, 104-107. There are several places throughout the manuscript where long sentences made the sections not only hard to read, but the key messages are lost in the weed.

>>>Thank you. We have edited the sentences on lines 100-102 and 104-107. We also made a full read-through of the manuscript with an eye toward reducing extra-long sentence length.

• I encourage you to avoid the use of certain words such as “Surprisingly,” Distressingly, “Absurd,” “Not surprisingly,” “Understandably,” “chilly climate,” etc. These words could through your readers off because they try to impose your personal views on the issues on your readers.

>>>We have removed all of the adverbs quoted above. We removed “chilly climate” from the abstract and replaced it with “less receptive climate.” We left “chilly climate,” as a descriptive term, in the Discussion, as we believe this metaphor is supported by our data.

• I suggest removing the word “Act of God” and use “natural disaster” for the same reason I mention in my preceding comments.

>>>We have removed “Act of God.” 

• Please clarify how cultural stigma offset cultural preference.

>>>With apologies, we do not understand this request for clarification.

• Please also clarify how media, the internet, etc. are cultural products or define what you meant by cultural products.

>>>In the sentence in question (pasted below in quotations), we offer media, internet content, and advertising as face-valid exemplars of cultural products, but refer interested readers to the Lamoreaux and Morling article cited in parentheses for more detail on cultural products.

“Individuals and institutions (Delker, 2020) and cultural products (e.g., media, internet content, advertising; Lamoreaux & Morling, 2013) can be vehicles of cultural stigma.”

• I would change the word “degradation” - line 208.

>>>Writing about trauma, experiences that by their nature can be ‘unspeakable,’ is challenging, and we appreciate the close attention to our word choice. In this case, we have retained the word “degradation” because we believe it descriptively evokes the experience of being the victim of interpersonal violence committed intentionally; to ‘degrade’ is “treat or regard (someone) with contempt or disrespect.” For readers who may be unfamiliar with the felt sense of interpersonal violence, we believe the words ‘degrade’ or ‘degradation’ captures some of the insult to self-worth than can be attached to interpersonal violence.

• The sentence on line 217-218 “ The perception that survivors of interpersonal violence must bear the burden to tell their stories is consistent with the implicit cultural assumption that victims are personally responsible for what happens to them,” contradict with eh what is stated on line 161-162 regarding the invisible nature of the interpersonal violence.

>>>Yes, we agree whole-heartedly that these are paradoxical statements. In the sentences after the sentence quoted from lines 217-218, we propose that “there is a paradoxical set of dominant cultural expectations for stigmatizing trauma survivors: to be silent about their experiences and to tell others about their experiences, if they can tell a story with a happy, and empowered, ending.” This cultural paradox is part of what we are testing with our hypotheses in this study.

Method

• Something is off on line 287 regarding average age and age range.

>>>Thank you. We erroneously stated that the overall average age was “mid-to-late 20s.” It was actually late 20s and we have fixed that. In examining the data tables to address this point of feedback, we also realized that the sample size for our Qualtrics sample was erroneously listed as 315; it was actually 190. We corrected this error in the Method section, Table 1, and in the total N listed in the Abstract (now N = 1,872).

• Please review your data in Table 1. There several issues with the percentage values.

>>> We have added a clarifying sentence to the note at the bottom of the table: “Total n(%) of race items do not add up to 100% because participants could select more than one.”

• Your participants were self-selected. You may want to explain the impact of self-selection bias on your finding somewhere in your manuscript, preferably in the limitation section.

>>>We removed the term “self-selection” from the procedure, as this is a misnomer. Participants did not self-select into the study to a degree greater than any participant chooses to participate in a study. They chose to participate in the study based on the title of the study, “Evaluating Stories.”

• Lined 297: I would use “incentive” instead of “compensation” unless you fully compensated the participant based on the locally approved hourly wage.

>>>We have changed “compensation” to “incentive.”

• Do your participants, especially university students, know each other? How that defies the purpose of the random assignment? What measures were taken to reduce the risks? How might that have affected your results?

>>>Some of our university (not MTurk or Qualtrics) participants may have known each other, but would not necessarily know that each other had participated in the study, as participation was confidential and anonymous. We are uncertain as to how the results would be impacted by student knowledge of each other, especially when participants did not know our hypotheses and when the words “redemption” and “stigma” were never used in the study materials.

Discussion/Conclusion

• It appears that you made a very strong generalization. I do not think you nationally representative data to make an inference to the general US population.

>>>We feel confident generalizing to the U.S. population based on the data we have, nearly 2000 adults from a range of geographic regions in the U.S. and representing various genders and ethnicities (though like the U.S. population, over-representing White individuals). However, the question of the extent to which our findings generalize to various populations is an important one. We offer a number of suggestions for future research in this regard in the Limitations and Future Directions section.

• There are situations where individuals are legally obligated to tell the act of violence. Your sentence on line 646 may need to reflect circumstances where individuals have a legal obligation to tell the story.

>>>There may be situations where individuals are legally obligated to tell the act of sexual violence, but the situations that first come to our minds are situations where these stories are not told in court because victims do not wish to have their characters publicly attacked by the prosecution, to face perpetrators in court, or to testify against intimate partners. The sentence in question (formerly on line 646) states, generally, that it is not desirable to force victims to tell their stories (without commenting on the legal aspects).

• I do not think your findings reflect that the US society has deep ambivalence and distorted thinking about interpersonal violence. This is really making a robust inference to the general population which I do not believe you have enough statistical power to make that generalization.

>>>We have removed the phrase “distorted thinking” and the qualifier “deep.” We believe that the statement about public ambivalence is supported by our data that the public recognizes that it is more difficult and less likely to tell sexual violence stories, and yet, sexual assault victims are questioned as to why they do not say something sooner, or at all (Stewart, 2019). 

• You suggest that professionals can be encouraged to recognize the difficulty of sharing a story of interpersonal violence and offer validation around this difficulty, but you did not offer action steps. Also, since most of the victimizations happen in the community among the margin lied group (e.g., domestic workers), what other recommendations you give to those working to help the most vulnerable group?

>>>The question of how to support those professionals working with the most vulnerable groups is an important one and lies beyond the scope of what we feel qualified to advise, based on our data. We added the following acknowledgement of the barriers to trauma disclosure for acutely vulnerable groups, a question that we deal with more extensively in a theory paper cited in the manuscript (Delker, Salton, & McLean, 2019).

“Advocates and scholars have drawn attention to the systemic and societal barriers to sexual violence story-telling for highly stigmatized, marginalized, and potentially isolated survivors, such as people with disabilities (Cramer, Plummer, & Ross, 2019) and immigrants (who may or may not have limited-English proficiency, and be undocumented or part of a “mixed status” immigrant family; Huang, 2019).”

Reviewer #2: 

This was an excellent paper, congratulations. It was very well written, intellectually stimulating, conceptually mature, and highly significant. The impact for the field is huge. I might also recommend reading this article: https://medium.com/@p.sawrikar/hypocritical-wiring-and-its-limits-on-empathy-the-sense-of-agency-bias-9b9aeaaba3b3?source=friends_link&sk=774be8c483b91140dadedf988f889792

 >>>Thank you for your praise for our work, and for recommending this article!

6. PLOS authors have the option to publish the peer review history of their article (what does this mean?). If published, this will include your full peer review and any attached files.

Do you want your identity to be public for this peer review? For information about this choice, including consent withdrawal, please see our Privacy Policy.

Reviewer #1: No

Reviewer #2: No

---

## [Decision Letter · Decision Letter 1]

21 May 2020

Who has to tell their trauma story and how hard will it be? Influence of cultural stigma and narrative redemption on the storying of sexual violence

PONE-D-20-02734R1

Dear Dr. Delker,

We are pleased to inform you that your manuscript has been judged scientifically suitable for publication and will be formally accepted for publication once it complies with all outstanding technical requirements.

With kind regards,

Whitney S. Rice, DrPH

Academic Editor

PLOS ONE

Additional Editor Comments (optional):

Reviewers' comments:

Reviewer's Responses to Questions

**Comments to the Author**

1. If the authors have adequately addressed your comments raised in a previous round of review and you feel that this manuscript is now acceptable for publication, you may indicate that here to bypass the “Comments to the Author” section, enter your conflict of interest statement in the “Confidential to Editor” section, and submit your "Accept" recommendation.

Reviewer #1: All comments have been addressed

2. Is the manuscript technically sound, and do the data support the conclusions?

Reviewer #1: Yes

3. Has the statistical analysis been performed appropriately and rigorously? 

Reviewer #1: Yes

4. Have the authors made all data underlying the findings in their manuscript fully available?

Reviewer #1: Yes

5. Is the manuscript presented in an intelligible fashion and written in standard English?

Reviewer #1: Yes

6. Review Comments to the Author

Reviewer #1: Authors did great revising this manuscript, clarifying their views and making changes to some critical areas that needed adjustments. After reviewing it again, I see the document will add high scientific and practical values. The only minor discomfort I have with this paper is the use of the word redemption. When referring to the theoretical framework of redemption story-telling, the authors chose to take a very simplistic approach to the U.S. cultural history or how redemptive stories among U.S. audiences would be understood. Their view is simplistic and relies on one school of thought, and it does not recognize that the U.S. audience is made of the world population. They responded, saying they regret if any reader perceived a tone of diminishment. In my view, this again dismisses the broad U.S. audience that may perceive the authors' assertion as diminishing the role, bravery, and courage of the victims in overcoming adversity. The original comment is not dismissive of the authors' view, but it was meant to show them how other people would view the authors' assertion of this specific point and to give the authors a chance to look at it from a different angle.

7. PLOS authors have the option to publish the peer review history of their article (what does this mean?). If published, this will include your full peer review and any attached files.

Reviewer #1: No

---

## [Editor Report · Acceptance letter]

28 May 2020

PONE-D-20-02734R1 

Who has to tell their trauma story and how hard will it be? Influence of cultural stigma and narrative redemption on the storying of sexual violence 

Dear Dr. Delker:

I am pleased to inform you that your manuscript has been deemed suitable for publication in PLOS ONE. Congratulations! Your manuscript is now with our production department. 

With kind regards,

on behalf of

Dr. Whitney S. Rice 

Academic Editor

PLOS ONE